# Health literacy and associated factors among undergraduates: A university-based cross-sectional study in Nepal

Sandesh Bhusal[1,2], Rajan Paudel[1], Milan Gaihre[3], Kiran Paudel[1,2], Tara Ballav Adhikari[2,4,5], Pranil Man Singh Pradhan[6,7] *

1 Institute of Medicine, Tribhuvan University, Kathmandu, Nepal, 2 NCD Watch Nepal, Maharajgunj, Kathmandu, Nepal, 3 College of Medical Sciences, Kathmandu University, Chitwan, Nepal, 4 Nepal Development Society, Bharatpur, Chitwan, Nepal, 5 Center for Global Health, Department of Public Health, Aarhus University, Aarhus, Denmark, 6 Department of Community Medicine, Maharajgunj Medical Campus, Institute of Medicine, Tribhuvan University, Kathmandu, Nepal, 7 Nepalese Society of Community Medicine, Kathmandu, Nepal

* pranil.pradhan@gmail.com

**Data Availability Statement:** All data are available in the Supporting information files.

**Funding:** This study was supported by Nepal Health Research Council (NHRC), Nepal under

## Abstract

Health literacy is one of the most critical aspects of health promotion. Limited health literacy is also accounted for adverse health outcomes and a huge financial burden on society. However, a gap exists in the level of health literacy, especially among undergraduates. This study aimed to assess the levels of health literacy and its socio-demographic determinants among undergraduate students of Tribhuvan University, Nepal. A web-based cross-sectional survey was conducted among 469 undergraduate students from five institutes of Tribhuvan University, Nepal. The 16-item short version of the European Health Literacy Survey Questionnaire (HLS-EU-Q16) was used to measure students' health literacy levels. Associated factors were examined using Chi-square tests followed by multivariate logistic regression analyses at the level of significance of 0.05. Nearly 61% of students were found to have limited health literacy (24.5% had "inadequate" and 36.3% had "problematic" health literacy). Female students (aOR = 1.6, 95% CI: 1.1–2.5), students from non-health related majors (aOR = 1.9, 95% CI: 1.2–3.0), students with unsatisfactory health status (aOR = 2.8, 95% CI: 1.7–4.5), students with poor financial status (aOR = 2.9, 95% CI: 1.2–6.8) and students with low self-esteem (aOR = 2.5, 95% CI: 1.5–4.1) were significantly more likely to have limited health literacy. The majority of the undergraduates were found to have limited health literacy. Gender, sector of study, self-rated health status, self-rated financial status, and self-esteem were significantly associated with limited health literacy. This study indicates university students should not be assumed to be health-literate and interventions to improve students' health literacy especially for those whose majors are not health-related should be implemented. Further studies using a longer version of the health literacy survey questionnaire and qualitative methods to explore more on determinants of health literacy are recommended.

Undergraduate Health Research Grant Program 2021 (Ref. no. 2824). SB was the recipient of this grant. The funders had no role in study design, data collection and analysis, decision to publish, or preparation of the manuscript.

**Competing interests:** Authors declare no competing interests.

**Abbreviations:** CI, Confidence Interval; HLS, Health Literacy Survey; NCDs, Non-Communicable Diseases; OR, Odds Ratio; SD, Standard Deviation; SDG, Sustainable Development Goals; UHC, Universal Health Coverage.

## Introduction

Health literacy represents the cognitive and social skills that determine the motivation and ability of individuals to gain access to, understand, and use information in ways that promote and maintain good health [1]. The World Health Organization (WHO) has positioned health literacy as a key mechanism to meet the health-related Sustainable Development Goal (SDG3) [2]. It has recently been shown to be important for improving Universal Health Coverage (UHC). Improving UHC must not only focus on providing infrastructure but also equipping people to be able to explore, understand, and use existing channels to enhance their health [3]. However, there is little known about the status of health literacy in developing countries such as Nepal.

People sometimes have difficulties in seeking health care, understanding health information, difficulty communicating with health care professionals, adherence to medical regimens, etc. which have a potential role in self-care and management of chronic diseases [4]. Limited health literacy is associated with adverse health outcomes, health inequalities, and a huge financial burden on society [5, 6]. Health literacy has been shown as a stronger predictor of health status than any other socioeconomic factors [7, 8]. According to a recent meta-analysis, health literacy was moderately correlated with quality of life [9]. Therefore, improving the level of health literacy should be put forward as an essential action for promoting health.

Nepal has a unique landscape of rich culture and traditional beliefs and practices which have implications for health and diseases. There are still many misconceptions and deep-rooted cultural beliefs about health, illness, and the healthcare system [10, 11]. It is relevant to measure health literacy coupled with these cultural factors as there are very few studies conducted to examine health literacy in Nepal [12].

The undergraduate stage is regarded as a stage with the greatest learning potential and this stage is critical for forming the framework of health literacy [8]. For many students, the university is a period of transition from teenager to young adult, moving out of home and relying less on parents to make health-related decisions [13]. Understanding the health literacy levels of these younger populations and then addressing any gaps provides a mechanism towards producing health literate professionals who can understand and respond to the health literacy needs of the families and communities [14, 15]. This population constitutes a major proportion and is crucial for the success of any health related promotion or prevention efforts. They are very receptive to information, and therefore, healthy behavior established at this phase of life is more likely to be continued [16].

University students may reasonably be expected to demonstrate good levels of health literacy; however, various studies worldwide have demonstrated poor health literacy among undergraduates [8, 17]. So, it is essential to explore the level of health literacy among undergraduates. Therefore, this study aimed to assess the level of health literacy and factors determining limited health literacy among undergraduate students in Tribhuvan University, Nepal.

## Methods

### Study setting

The study was conducted at Tribhuvan University, Nepal. Tribhuvan University is a public, oldest, and the largest university in terms of students' enrollment in Nepal. It is also characterized by its diverse student bodies, which are representative of the Nepalese population.

All the five institutes of the university; Institute of Medicine (IOM), Institute of Engineering (IOE), Institute of Forestry (IOF), Institute of Agriculture and Animal Science (IAAS), and

Institute of Science and Technology (IOST), were selected as the study sites. Then, one campus from each of the institutes was purposively selected. Maharajgunj Medical Campus, Pulchowk Campus, Hetauda campus, Lamjung Campus, and Trichandra Multiple Campus were selected respectively from IOM, IOE, IOF, IAAS, and IOST to recruit the participants.

## Study population

The study population included the students pursuing an undergraduate degree in various majors across the five campuses of respective institutes of Tribhuvan University, Nepal. All undergraduate students from any of the five campuses were eligible for inclusion. Postgraduate students, international students, and students undertaking affiliated programs were excluded.

## Study design and sample

A web-based cross-sectional survey was conducted between January and February 2021. Information about students' enrollment and population size in each campus was obtained from the Dean's office and information officer of Tribhuvan University, Nepal. Responses were collected from each of the five campuses of respective institutes. The convenience sampling technique was employed to select participants.

The expected proportion in population was taken as 45% from a similar study conducted in Ghana [17] and the sample size was calculated using the formula; $n = z^2pq/e^2$ where $p = 0.45$, $q = 0.55$, $z = 1.96$ at 95% confidence interval, and $e = 0.05$. Assuming 25% as the non-response rate, the sample size of 475 was determined.

## Data collection

A web-based survey approach was taken to collect data from participants and Google forms were administered via e-mail. Single response from each student was ensured via Google Forms setting by choosing 'Limit to 1 response'.

## Measurements

**Dependent variable** *(Health literacy).* English version of the European Health Literacy Survey Questionnaire with 16 items (HLS-EU-Q16) was used to assess health literacy [18, 19]. This measure focuses on perceived difficulties/ease in accessing, understanding, appraising, and applying health information across the domains of health care, disease prevention, and health promotion [20].

The 16 items have four responses (very easy, easy, difficult, and very difficult) with a "don't know" option. All responses were given a numerical code as follows: 1, very difficult; 2, difficult; 3, easy; 4, very easy; and 0, don't know. The mean score was calculated for all items on the scale, and then it was converted to an index using the formula below per recommendations of the European health literacy consortium.

*Health literacy index score = (mean—1) * (50/3)*, where mean is the average of items on the scale, 1 = the minimal value of the mean, 3 = the range of the mean, and 50 = the chosen maximum value of the new index scores [18, 21].

The index scores were recoded into four health literacy categories as per the threshold established by HLS-EU consortium: *excellent (>42–50); sufficient (>33–42); problematic (>25–33); and inadequate (0–25).* Later, health literacy categories were dichotomized into limited (inadequate and problematic health literacy categories combined) and adequate health literacy (sufficient and excellent health literacy categories combined). During the pretesting no

one reported the difficulties in understanding the questionnaire and we decided to administer questionnaire in English version.

**Independent variables.** Socio-demographic factors included age, sex, place of origin (categorized as rural and urban), family type (nuclear, joint and extended), highest educational level of parents (illiterate/no formal schooling, below secondary school level [< 10 years of formal education], secondary school and above [≥10 years of formal education] and ethnicity (categorized as Brahmin/Chhetri, Janajati, Madhesi, Muslim, Dalit and Others according to Health Management Information System, Nepal government) [22]. The academic year of the students was categorized into "lower year" and "higher year', where the 1st or 2nd year of the study was regarded as lower year and ≥3rd year as the higher year [17, 23].

Other variables were self-rated financial status, self-rated health status, and self-rated self-esteem. Self-rated financial status was measured on a scale of 1 to 6 which included the responses from very poor to very rich [17], categorized as a new variable as: 1–2 (poor) and 3–6 (good). Self-rated health status was classified as excellent, good, moderate, poor, and very poor. This was re-categorized into satisfactory health and unsatisfactory health status. Responses "excellent" and "good health" were combined into satisfactory, while all other responses were categorized as unsatisfactory health status. Self-rated self-esteem was measured on a scale of 1 to 7 using the Rosenberg scale [24]. Measurement on the scale was based on response to the item "I have very high self-esteem" where the lowest value represents the "not very true of me" and the highest value "very true of me".

## Data analyses

Statistical analysis was performed using IBM SPSS version 20. Descriptive analysis was done to identify the distribution of socio-demographic characteristics of participants. Association between independent variables and health literacy score category was measured by Chi-Square tests followed by binary logistic regression analyses. The level of statistical significance was considered to be $p < 0.05$.

## Ethical considerations

The approval was obtained from the Institutional Review Committee of the Institute of Medicine (IOM), Tribhuvan University, Nepal (Reference no; 1561(6–11) E2/077/078). All the respondents were informed about the aims and objectives of the study by including the written consent form in the questionnaire itself. Written digital consent was taken from study participants prior to completing the survey form. The research ethics committee waived the need for consent from guardians of minors included in the study. Participants gave their consent by ticking the designated box.

## Results

Table 1 presents the background information of the respondents.

## Socio-demographic characteristics

A total of 469 respondents were included in the final analysis. We did not receive complete response from six participants. The mean age (± SD) of the respondents was 20.9 (± 1.7) years. Among the respondents, the majority were males (54.1%). Brahmin/Chhetri (75.7%) was the major ethnic group followed by Janajati (17.3%) and Madhesi (5.1%). While categorizing place of origin as rural and urban, the majority of respondents (60.6%) were from rural settlements. 71.9% of the participants belonged to a nuclear family.

**Table 1. Characteristics of the participants (n = 469).**

| Characteristics | Number (%) |
|---|---:|
| **Age (in years)** | |
| < 20 years | 76 (16.2) |
| ≥ 20 years | 393 (83.8) |
| Mean ± SD | 20.9 ± 1.7 |
| **Gender** | |
| Female | 214 (45.6) |
| Male | 255 (54.4) |
| **Ethnicity** | |
| Brahmin/Chhetri | 355 (75.7) |
| Janajati | 81 (17.3) |
| Madhesi | 24 (5.1) |
| Others | 9 (1.9) |
| **Place of origin** | |
| Rural | 284 (60.6) |
| Urban | 185 (39.4) |
| **Institutes** | |
| IOM | 124 (26.4) |
| IOE | 165 (35.0) |
| IOF | 60 (12.8) |
| IAAS | 63 (13.4) |
| IOST | 58 (12.4) |
| **Student's sector** | |
| Non-health | 345 (73.6) |
| Health | 124 (26.4) |
| **Academic year** | |
| First | 94 (20) |
| Second | 170 (36.2) |
| Third | 112 (23.9) |
| Fourth and above | 95 (19.9) |
| **Family type** | |
| Nuclear | 337 (71.9) |
| Joint/Extended | 132 (28.1) |
| **Father's education** | |
| Illiterate/No formal schooling | 41 (8.7) |
| Below secondary | 136 (29) |
| Secondary and above | 292 (62.3) |
| **Mother's education** | |
| Illiterate/No formal schooling | 111 (23.7) |
| Below secondary | 193 (41.2) |
| Secondary and above | 165 (35.2) |

About 73.6% of the students were from non-health-related faculties while 26.4% were from the directly health-related faculties as the Institute of Medicine (IOM) is the only institute running health programs among five institutes of Tribhuvan University. Talking about the parent's highest level of education, 62.3% of fathers had attained education of secondary level or above while only 35.2% mothers had attained that level of education.

**Table 2. Health literacy by health domains and health competencies (n = 469).**

|  | HLS-EU-Q16 Items | Mean (SD) |
|---|---|---|
| **Health competencies** |  |  |
| Access | 4 | 2.70 (0.57) |
| Understand | 6 | 3.03 (0.52) |
| Appraise | 3 | 2.52 (0.70) |
| Apply | 3 | 2.84 (0.60) |
| **Health Domains** |  |  |
| Health Care | 7 | 2.85 (0.53) |
| Disease prevention | 5 | 2.74 (0.63) |
| Health promotion | 4 | 2.86 (0.55) |

## Health literacy by health domains and health competencies

Table 2 shows that, when analyzing the participant's performance based on four health literacy competencies, students scored high on competency dealing with understanding health information [Mean (SD): 3.03(0.52)] while they scored low on competency dealing with appraising health information [Mean (SD): 2.52 (0.70)].

When comparing all competencies over the domains of healthcare, disease prevention, and health promotion, the mean score per item was highest within the domain of health promotion [Mean (SD): 2.86 (0.55)] and lowest in the domain of disease prevention [Mean (SD): 2.74 (0.63)].

## Factors associated with limited health literacy

About 61% of the undergraduate students were found to have limited health literacy. Female students were 1.6 (95% CI: 1.1–2.5) times more likely to have limited health literacy as compared to males. Students whose majors were not directly related to health were almost twice more likely to have limited health literacy than students of health-related majors (aOR = 1.9, 95% CI: 1.2–3.0). Students who perceived their health status as unsatisfactory had higher odds of limited health literacy compared to students who reported having satisfactory health status (aOR = 2.8, 95% CI: 1.7–4.5). Compared to students who have good financial status, students with low financial status were 2.9 times more likely to have limited health literacy (aOR = 2.9, 95% CI: 1.2–6.8). Students having low self-esteem were more likely to have limited health literacy than those who have high self-esteem (aOR = 2.5, 95% CI: 1.5–4.1) (Table 3).

## Discussion

The study found that 60.8% of the undergraduates studying at Tribhuvan University, Nepal, had limited health literacy. Our study finding was consistent with a Ghanaian university-based study [17]. The population based health literacy survey conducted in eight European countries showed 47% had limited (insufficient or problematic) health literacy [21]. However university-based studies, especially in the United States and Canada, have reported better levels of health literacy (about 7%-15% limited health literacy) [25, 26]. These huge differences might be due to difference in study settings as these countries are highly developed and richer than Nepal. Economic development levels, health resource allocations, and access to health information are lower in developing countries like Nepal compared to developed countries like the United States and Canada.

**Table 3. Factors associated with limited health literacy (n = 469).**

| Characteristics | Limited health literacy number (%) | Unadjusted odds ratio (95% CI) | Adjusted odds ratio (95% CI) |
|---|---|---|---|
| **Age (in years)** | | | |
| ≥ 20 years (ref) | 233 (59.3) | | |
| < 20 years | 52 (68.4) | 1.5 (0.9–2.5) | 1.4 (0.8–2.5) |
| **Gender** | | | |
| Male (ref) | 145 (56.9) | | |
| Female | 140 (65.4) | 1.4 (0.9–2.1) | 1.6 (1.1–2.5)* |
| **Ethnicity** | | | |
| Brahmin/Chhetri (ref) | 208 (58.6) | | |
| Non-Brahmin/Chhetri | 77 (67.5) | 1.5 (0.9–2.3) | 1.4 (0.8–2.3) |
| **Place of origin** | | | |
| Urban (ref) | 113 (61.1) | | |
| Rural | 172 (60.6) | 1.0 (0.7–1.5) | 1.0 (0.7–1.6) |
| **Student's sector** | | | |
| Health (ref) | 63 (50.8) | | |
| Non-health | 222 (64.3) | 1.7 (1.2–2.6)* | 1.9 (1.2–3.0)* |
| **Academic year** | | | |
| Higher year (ref) | 120 (58.5) | | |
| Lower year | 165 (62.5) | 1.2 (0.8–1.7) | 0.9 (0.6–1.5) |
| **Family type** | | | |
| Nuclear (ref) | 206 (61.1) | | |
| Joint/Extended | 79 (59.8) | 0.9 (0.6–1.4) | 0.7 (0.5–1.2) |
| **Father's education** | | | |
| Illiterate/No formal schooling (ref) | 26 (63.4) | | |
| Below secondary | 84 (61.8) | 1.2 (0.6–2.3) | 0.7 (0.3–1.6) |
| Secondary and above | 175 (59.9) | 1.1 (0.7–1.6) | 1.0 (0.6–1.7) |
| **Mother's education** | | | |
| Illiterate/No formal schooling (ref) | 70 (63.1) | | |
| Below secondary | 117 (60.6) | 1.2 (0.7–1.9) | 1.0 (0.5–1.9) |
| Secondary and above | 98 (59.4) | 1.1 (0.7–1.6) | 1.0 (0.6–1.7) |
| **Self-rated health status** | | | |
| Satisfactory (ref) | 172 (52.8) | | |
| Unsatisfactory | 113 (79) | 3.4 (2.1–5.3)** | 2.8 (1.7–4.5)** |
| **Self-rated financial status** | | | |
| Good (ref) | 247 (58.4) | | |
| Poor | 38 (82.6) | 3.4 (1.5–7.4)** | 2.9 (1.2–6.8)* |
| **Self-rated self esteem** | | | |
| High (ref) | 182 (54) | | |
| Low | 103 (78) | 3.0 (1.9–4.8)** | 2.5 (1.5–4.1)** |

* indicates significance at $p$-value < 0.05,

** indicates significance at p-value < 0.001

In this study, females were found to be 1.6 times more likely to have limited health literacy than males. This contrasts with the findings from a similar study conducted among Danish adults [27] and university students in Turkey [28]. Patriarchal societies where households tend to favor males for healthcare services, variations in the educational systems and the other sociocultural characteristics may have attributed to this discrepancy.

Higher odds of having limited health literacy were found on the students from non-health related majors as compared with students from health-related majors. Similar differences were seen in other studies conducted in countries like Ghana [17], China [8], Jordan [23] and America [25]. The possible explanation could be students in health-related programs familiarity with the health-related knowledge, the health-care environment, topics of health promotion, and disease prevention.

Students from rural origin reported a lower level of health literacy than students from an urban origin in a study from China [8]. However, in our study no significant association was found between health literacy and place of origin. It might be due to the difference in sampling size. Students who reported their health status as unsatisfactory were more likely to have a lower level of health literacy, consistent with existing literatures [29–31]. This pattern is likely due to their difficulty in navigating the healthcare system and possessing insufficient health information for self-care.

Individuals who perceived their financial status as poor had higher odds of having limited health literacy compared to their counterparts. This is consistent with the results of some studies conducted among university students [7, 17, 27]. Lower economic status impacts the access, use, and quality of health care. This factor might have played a role in health literacy skills of students. In our study, students with high self-esteem levels had higher health literacy than their counterparts who rated themselves as having low self-esteem. Adolescents' mental and physical health status were found to be associated self-esteem in a study conducted in New Zealand [32]. This evidence might explain the role of self-esteem in determining the level of health literacy.

Contrary to our expectation, health literacy was not associated with the academic year and parent's highest level of education as reported in previous studies [7, 8, 14, 17, 33]. This absence of association is not well understood and needs to be explored prospectively in future research. In our study, the mean score per item (over all competencies) was lowest for appraising health information and while comparing all competencies over the three domains, the mean score per item was highest within the domain of health promotion. Therefore, our findings corroborate with findings from study conducted in Denmark [27].

## Strengths and limitations

To the best of our knowledge this is the first study to explore health literacy among university undergraduates and compare the health literacy levels among health and non health- related students in Nepal. The study provided information on student's self-perceived competencies necessary for them to make informed health decisions. The findings of this study added evidence into the limited literature on the health literacy level in Nepal.

The present study had few limitations. All the measurements in this study were based on self-reports, which may have been prone to response and information bias. This study was cross-sectional and, therefore, cannot demonstrate causality between the factors associated with health literacy. As most of the participants were from non-health background, which might have created the discrepancy while analyzing the health literacy level in terms of health-related students versus non-health related students. Restrictions due to the COVID-19 pandemic limited our plans for on-site data collection so we had to collect the data online. Since this was a web-based study, limited access to the internet may have discouraged the students from participating in the survey.

## Conclusion

The study revealed a high prevalence of limited health literacy among university students in Nepal. Gender, sector of study, perceived health status, financial status and self-esteem were significantly associated with limited health literacy. According to the findings, even educated people, such as undergraduates, face difficulties interacting with health-care procedures and systems. University students should not be assumed to be health-literate and interventions that will help enhance their literacy in health should be implemented especially among non-health related institutions. Specific policy to make health literacy friendly health institutions must be implemented. Further studies with better sampling procedure using a longer version of health literacy survey questionnaire and qualitative methods to explore more on determinants of health literacy are recommended.

## Supporting information

**S1 Data.**
(XLSX)

**S1 Tool. Study questionnaire.**
(DOCX)

## Acknowledgments

We thank all the faculty members at the Department of Community Medicine and Central Department of Public Health, Institute of Medicine, Tribhuvan University, Nepal for their guidance during the research project. Our appreciation goes to individuals responding to the questionnaire and the class monitors for their communication and coordination.

## Author Contributions

**Conceptualization:** Sandesh Bhusal, Rajan Paudel, Pranil Man Singh Pradhan.

**Data curation:** Sandesh Bhusal, Rajan Paudel, Milan Gaihre, Kiran Paudel, Tara Ballav Adhikari.

**Formal analysis:** Sandesh Bhusal, Rajan Paudel, Tara Ballav Adhikari.

**Funding acquisition:** Sandesh Bhusal.

**Investigation:** Sandesh Bhusal.

**Methodology:** Sandesh Bhusal, Rajan Paudel, Pranil Man Singh Pradhan.

**Project administration:** Sandesh Bhusal, Pranil Man Singh Pradhan.

**Resources:** Sandesh Bhusal.

**Software:** Sandesh Bhusal.

**Supervision:** Sandesh Bhusal, Pranil Man Singh Pradhan.

**Validation:** Sandesh Bhusal.

**Writing – original draft:** Sandesh Bhusal, Rajan Paudel, Milan Gaihre, Kiran Paudel, Tara Ballav Adhikari, Pranil Man Singh Pradhan.

**Writing – review & editing:** Pranil Man Singh Pradhan.

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
