## [Decision Letter · Decision Letter 0]

9 Aug 2021

PGPH-D-21-00095

Health Literacy and Associated Factors among University Students in Nepal

Dear Professor Pranil Man Singh Pradhan,

Thank you for submitting the manuscript "Health literacy and associated factors among university students in Nepal" for consideration to PLOS Global Public Health. After careful consideration, we feel that it has merit but does not fully meet PLOS Global Public Health’s publication criteria as it currently stands. Therefore, we invite you to submit a revised version of the manuscript that addresses the points raised during the review process.

Both authors reflect on the need of improving the methods section. More information is thus needed to understand various proceeding associated with the online survey. As editor my main concern is also methodological. Authors should let the readers know about the population (averages of age, female proportion, parents’ education etc.) from which the study sample is drawn from. Otherwise the study may not show if sample biases have been corrected at all. The sample can be representative of maybe one or two students’ characteristics, and this should be communicated and adjusted when discussing results.

We look forward to receiving your revised manuscript.

Kind regards,

Jose Ignacio Nazif-Munoz, Ph.D.

Academic Editor

Journal Requirements:

Reviewer 1

Thank you for sharing the manuscript. It is a well-written manuscript. It has described the introduction, methods, results, and discussion well and these are aligned to the stated objectives.

There are few observations that the authors may consider.

Methods:

1. Method is well written and explained the tools adequately. But a brief description of the web-based survey tool, the language in which it was delivered would be welcome. Did all the tools [health literacy scale, financial, self-esteem and health status ] were put in the online survey tool, and whether it was delivered in English or Nepali Language may help the readers to understand the data collection process. A line or two on the consent process should be considered.

2. A brief description of the self-reported financial scale or how the information was collected will give the readers better insight into the analysis. Otherwise, attachment tools for financial status or health status can be considered as supplementary documents as per the journal guidelines can be considered.

Results:

3. Table 2: Table 2 has been described as, 'When comparing all competencies over the domains of healthcare, disease prevention, and health promotion, the mean score per item was highest within the domain of health promotion [Mean (SD): 2.86 (0.55)] and lowest in the domain of disease prevention [Mean (SD): 2.86 (0.63)]'.

But in table 2, the Mean SD of the disease prevention is 2.74 and not 2.86. The authors should check this discrepancy.

Discussion:

4. The paragraph above the strength and limitation where the authors refer to similarity with a study in Denmark [Ref 18]. It would be better if the authors briefly describe the similarity that they are trying to point.

Generic

5.The lack of page numbers and line numbers is a bit problematic for reviewing.

Reviewer 2

The manuscript submitted by Mr. Pranil Man Singh Pradhan and his team is important and has tried to fill the gap of literature in the area of health literacy. I enjoyed reading this work, and I have found some minor comments as well. Which I hope, the author would include into his work. I have embedded my comments by specifying each section of the manuscript.

Topic

1. Authors could make the topic more specific. Since, this study has been done in only one university of Nepal, it would not be fair to say the study among the university students. The current topic gives an idea to the readers that a study is among the university students in Nepal which, in fact, is not.

Abstract

1. The author has used HLS-EC-Q16 as a key word, but I did not find this abbreviation used anywhere in the abstract. I would suggest using the abbreviation first, at least once, and than use it in the keyword section.

Introduction

1. Not enough literature review has been done. How can limited health literacy leads to the economical burden and health inequalities in the society, you need to support this argument with evidence since you have mentioned this twice (in the abstract and second paragraph of an introduction).

2. The question like why this study was done in Nepal or how this study is relevant to Nepalese context is crucial. What are those believes, practices and mis-conception prevalent is Nepalese society which are preventing people to get informed about health information or have implication on health and disease in Nepal. Why was it so important to conduct this study in Nepal? Why you choose undergraduate student in Nepal? How this group (undergraduate) is so important in the Nepalese context that compel you to conduct this study? For example, did you choose this study group because this group represents the larger fraction of Nepalese population or because this group can influence the larger percentage of Nepalese population or because of something else? You need to work more on this. By only saying that undergraduate is a transition period, and it is always misunderstood as an educated group is enough to conduct this study. I think, enough studies need to be examined to answer these questions. The relation between three: Health Literacy, Nepal and Undergraduate students should be strong to carry out this study. This convinces others as well to make another study, expand this work and to contribute more to the area. When your rational to conduct a study in any region is strong, being an international reader  who do not know about Nepal, could understand your work more clarity and easily correlate the situation. It also makes an international reader to read a paper with the eyes of clarity.

3. In the 4th paragraph, the author has aimed to assess the level of comprehensive health literacy through this study. What does the word “comprehensive” means here? Does this questionnaire claim to measure the level of comprehensive health literacy? If yes, please state the word comprehensive with the evidence from claim.

Methodology:

1. I think, the author has not explained the inclusion and exclusion criteria very well which have created a confusion of understanding. The authors say that all undergraduates’ students are included. But I guess there must be international students in IoM, IoE or other institute pursuing their undergraduate degree. What about them? Did you include them or not? If yes, do they represent Nepal? Please make this clear.

2. The author has excluded the postgraduates’ students and affiliated programs. But what about the constituent campuses in IoM. There are 7 constituents’ campuses and 1 central department of public health under the IoM. Did you include the students from those constituents’ campuses? Are there any other constituent campuses under other institutes like the Institute of Medicine have? You should mention these thoughts very clearly. So that the readers should not have any confusion.

Tribhuvan university: http://tribhuvan-university.edu.np/institute/1_5db00959912d0#

3. The author has collected the sample from each institute proportionally based on the size of the population of each institute. So, what is the number of population/students enrolled in each institute in the corresponding year? How do I check whether they have collected the responses proportionally or not? How do I validate your sample taken from each institute? You must include the total number of students enrolled in each institute in the year in which study was pursued.

4. Did you translate the questionnaire into your local Nepalese language? If yes, how did you do that? Is the questionnaire tested before in the local language or the translated questionnaire have been piloted before? If you did not translate the questionnaire, did you receive any difficulties from the participants about the understanding of the words used in the questionnaire? Please make this very clear. This is important in many senses. One would be, it may give an idea to other researchers who wants to replicate or expand your work in the future by using the same questionnaire in Nepal.

5. The author has used a scale of 1 to 6 to measure self-rated financial status. Could you please clarify how did you build this scale? Is this an international standard? Is there any formula to develop this scale?

6. The author says limited access to the internet may have discouraged the students from participating in the survey. Does any student refused/denied participating in this survey due to internet problem or any other problem? If yes, what was the reason of denial. Was it internet or any cultural believes or any other reason? You should provide information regarding the refusal of participants if there was any.

7. Why did the author used shorter version of HLS-EC questionnaire? Why not the longer version of this questionnaire (HLS-EC Q16 vs Q47)?)? Please explain this. This may help you to identify strength or weakness of this study.

Results

1. The author in the Table 3 has categorized an academic year into Higher year and Lower year. What does higher and lower year mean? How did you classify this? Which year of students are included in higher and which groups are kept in lower year? Please state this.

2. Be careful about the correctness while writing the numbers and decimals. For example, you have used two values of mean (SD): 2.86 and 2.74 for disease prevention domain. Which one is correct? Please check the overall manuscript for such minor errors to convey the right idea.

Discussion and conclusion (Strengths and limitations)

1. There is huge difference in the sample size between health and non-health related students (26% vs 74%). Based on this sample difference, how can you say that the health-related students demonstrate higher level of health literacy? What if that was because of lower sample size of health-related students?

Reviewers' comments:

Reviewer's Responses to Questions

**Comments to the Author**

1. Does this manuscript meet PLOS Global Public Health’s publication criteria? Is the manuscript technically sound, and do the data support the conclusions? The manuscript must describe methodologically and ethically rigorous research with conclusions that are appropriately drawn based on the data presented.

Reviewer #1: Yes

Reviewer #2: Yes

2. Has the statistical analysis been performed appropriately and rigorously?

Reviewer #1: I don't know

Reviewer #2: Yes

3. Have the authors made all data underlying the findings in their manuscript fully available (please refer to the Data Availability Statement at the start of the manuscript PDF file)?

Reviewer #1: Yes

Reviewer #2: Yes

4. Is the manuscript presented in an intelligible fashion and written in standard English?

Reviewer #1: Yes

Reviewer #2: Yes

5. Review Comments to the Author

Please see above

6. PLOS authors have the option to publish the peer review history of their article (what does this mean?). If published, this will include your full peer review and any attached files.

**Do you want your identity to be public for this peer review?** For information about this choice, including consent withdrawal, please see our Privacy Policy.

Reviewer #1: No

Reviewer #2: No

---

## [Editor Report · Decision Letter 1]

18 Oct 2021

Health Literacy and Associated Factors among Undergraduates: A University-Based Cross-Sectional Study in Nepal

PGPH-D-21-00095R1

Dear Pr. Pranil Man Singh Pradhan

I am pleased to announce that I am recommending your work "Health Literacy and Associated Factors among Undergraduates: A University-Based Cross-Sectional Study in Nepal" for publication at Plos Global Public Health. You have considered observations and comments made by two independent reviewers, and this new version certainly has improved. In short we're pleased to inform you that your manuscript has been judged scientifically suitable for publication and will be formally accepted for publication once it meets all outstanding technical requirements.

Within one week, you'll receive an e-mail detailing the required amendments. When these have been addressed, you'll receive a formal acceptance letter and your manuscript will be scheduled for publication.

An invoice for payment will follow shortly after the formal acceptance. To ensure an efficient process, please log into Editorial Manager at https://www.editorialmanager.com/pgph/ click the 'Update My Information' link at the top of the page, and double check that your user information is up-to-date. If you have any billing related questions, please contact our Author Billing department directly at authorbilling@plos.org.

Thank you for having chosen our journal to disseminate your important research.

Warm regards,

Jose Ignacio Nazif-Munoz, Ph.D.

Academic Editor
